# Clean-label Backdoor Attacks by Selectively Poisoning with Limited Information from Target Class

**Quang H. Nguyen**[1]**, Nguyen Ngoc-Hieu**[1]**, The-Anh Ta**[2]**, Thanh Nguyen-Tang**[3]**,
Hoang Thanh-Tung**[4]**, Khoa D. Doan**[1]
[1]VinUniversity, [2]CSIRO's Data61, [3]Johns Hopkins University, [4]Vietnam National University
quang.nh@vinuni.edu.vn, ngochieutb13@gmail.com, theanh.ta@csiro.au,
nguyent@cs.jhu.edu, htt210@gmail.com, khoa.dd@vinuni.edu.vn

## Abstract

Deep neural networks have been shown to be vulnerable to backdoor attacks, in which the adversary manipulates the training dataset to mislead the model when the trigger appears, while it still behaves normally on benign data. Clean label attacks can succeed without modifying the semantic label of poisoned data, which are more stealthy but, on the other hand, are more challenging. To control the victim model, existing works focus on adding triggers to a random subset of the dataset, neglecting the fact that samples contribute unequally to the success of the attack and, therefore do not exploit the full potential of the backdoor. Some recent studies propose different strategies to select samples by recording the forgetting events or looking for hard samples with a supervised trained model. However, these methods require training and assume that the attacker has access to the whole labeled training set, which is not always the case in practice. In this work, we consider a more practical setting where the attacker only provides a subset of the dataset with the target label and has no knowledge of the victim model, and propose a method to select samples to poison more effectively. Our method takes advantage of pretrained self-supervised models, therefore incurs no extra computational cost for training, and can be applied to any victim model. Experiments on benchmark datasets illustrate the effectiveness of our strategy in improving clean-label backdoor attacks. Our strategy helps SIG reach 91% success rate with only 10% poisoning ratio.

## 1 Introduction

Modern deep learning models have exhibited tremendous success in solving challenging tasks, ranging from autonomous driving and face recognition to natural language processing. Training these large models requires massive training data, which is time-consuming, labor-intensive, and incurs huge costs to collect and annotate. Therefore, users usually prefer to employ third-party or open-source data. Recent studies have shown that deep models are vulnerable to backdoor attacks in which triggers are injected into the training process [10, 16, 9]. This poses a serious threat since a malicious data supplier can provide poisonous data such that the model trained on it behaves normally on benign data, but returns the output that the attacker wants when the trigger appears.

Most existing backdoor attacks rely on data poisoning and can be classified as either dirty-label or clean-label, depending on whether the label of poisoned data changes. For dirty-label attacks [10, 4, 22], the adversary adds trigger into data and *points* its label to their desired target label. Dirty-label backdoor attacks are effective but are easy to detect by humans during data verification since the semantics of the labels are changed. On the other hand, clean-label attacks [29, 1, 24] poison training data *without* changing the label making them more difficult to detect. However, compared to the dirty-label case, it is also much more difficult to mount clean-label backdoor attacks as one needs to poison more training data and the resulting models can have poor performance on clean data. In this

Published at NeurIPS 2023 Workshop on Backdoors in Deep Learning: The Good, the Bad, and the Ugly.

paper, we focus on improving the data effectiveness of backdoor attacks: specifically, to increase the attack performance given a small budget for the number of poisoned samples.

Prior backdoor attacks implicitly assume all training samples contributed equally to the attack success, and perform data poisoning uniformly randomly over training data. However, recent research [13, 12, 27] reveals that among training data points, some are more important while some others are redundant and can be discarded from the training set. One can ask a similar question for backdoor learning: *"Can selectively, rather than randomly, poisoning some training data points lead to more effective attacks?"* [31] and [8] studied this problem and proposed strategies to improve the efficiency in selecting samples to poison by recording forgetting events or looking at loss values to identify hard samples. However, these methods have several drawbacks. First, they need to train a model on the dataset from scratch, which is time-consuming and computationally expensive. Second, they require access to the whole training data to train the surrogate model, which is not applicable in some real-world scenarios where the user collects data from multiple sources, and the attacker has no knowledge of training data other than that it supplies.

Table 1: Properties of selection strategy. "✗" indicates the method lacks the property, "✓" indicates the method has the property.

| Method | trigger-agnostic | model-agnostic | no training required | partial data access |
|---|---|---|---|---|
| FUS [31] | ✗ | ✓ | ✗ | ✗ |
| Hard samples [8] | ✓ | ✓ | ✗ | ✗ |
| BASEL | ✓ | ✓ | ✓ | ✓ |

This paper considers a more practical setting where the attacker only provides data for the target class. We propose **B**ackdoor **A**ttacks by **SE**lective Poisoning with **L**imited Information (BASEL), a simple yet effective strategy for selecting samples to be used in clean-label attacks.

For the victim model to learn the backdoor, it needs to focus on the trigger rather than other features in the data [29]. Intuitively, if the samples with triggers are difficult to learn, the model will use triggers as shortcuts to minimize the objective function. As a result, the model is more prone to learning the triggers. To achieve such a goal without having access to the full training dataset or victim model, we propose a novel data selection strategy that uses self-supervised, pretrained models to identify hard training samples and add triggers to these samples. Using self-supervised, pretrained models makes our method (i) independent of the trigger and the victim model, (ii) does not require training, and (iii) does not require knowledge about other classes in the training set. Table 1 compares our method with existing methods.

In summary, our contributions include:

- BASEL, a new simple, model-agnostic, trigger-agnostic data selection strategy, which exploits pretrained models to select data points to poison for more successful clean-label backdoor attacks, without training or access to other classes.

- Extensive experiments on benchmark datasets to demonstrate the effectiveness of our method.

## 2 Related Works

### 2.1 Backdoor Attacks

Backdoor attacks aim to insert a malicious backdoor into the victim model. The first attempt is BadNets [10], where the attacker adds a predefined image patch to some images in the training set and changes the labels of these images to the target class. Follow-up works introduce various forms of the Trojan horse to enhance the stealthiness and the effectiveness of the attack, examples include blended [4], dynamic [25], warping-based [22], input-aware [21, 17], and learnable trigger [6]. These attacks are called dirty-label attacks as they change the true labels of poisoned examples.

Despite the success in manipulating the victim, dirty-label attacks can be easily spotted through human inspection. Clean-label backdoor attacks are attack methods that perverse the original labels of poisoned data points, and thus are more stealthy than dirty-label attacks. [29] suggested that using dirty-label attack triggers is ineffective for implementing clean-label attacks and proposed a data preprocessing method for implementing clean-label attacks. In the meantime, stronger triggers have been proposed. SIG [1] uses sinusoidal signals as backdoors. Refool [19] uses physical reflection models to implant reflection images into the dataset. HTBA [24] optimizes the input such that it

looks similar to the target label in the pixel space but close to the malicious image in the latent space. However, these attacks require a high poisoning rate and/or result in inferior success rates.

Research in backdoor attacks focuses on designing the trigger pattern, ignoring the possibility that benign samples chosen to attack can also play an important role. FUS [31] first showed that the number of forgetting events is an indicator of the contribution to the attack, and proposed a data selection strategy based on forgetting events that resulted in a better attack success rate. [8] identified three classical criteria to pick samples for clean-label attacks, namely loss value, gradient norm, and forgetting event. To select samples for poisoning, these methods require a surrogate model trained on a dataset with all training set classes, which is expensive and not always feasible.

## 2.2 Backdoor Defenses

Along with the emergence of backdoor attacks, defense methods to protect models are an active research area. Backdoor defenses can be categorized into two lines: backdoor detection and backdoor mitigation. Detection methods can be performed by examining the training dataset [23], verifying if the model is safe to deploy [3, 30], or spotting poisoned test samples [7]. Mitigation methods aim to reduce the backdoor effect in the model [18, 15]. From a security perspective, the adversary should not only succeed in attacking the model but also in dodging backdoor defenses.

## 3 Threat Model

In this section, we describe the threat model being studied in this paper.

**The attacker's goal.** The objective of the adversary is to inject a trigger into the victim model, such that the model acts normally on benign data, but misclassifies with the presence of the trigger. For instance, a facial recognition system that recognizes people to grant them certain permissions, but when being poisoned with sunglasses as a trigger, it will give full authority to anyone wearing sunglasses.

**The attacker's ability.** We focus on data-poisoning scenarios, where the attack poisons the dataset and supplies it to the victim. In the above example, to build a facial recognition model each person is asked to provide their photos. Malicious users can inject triggers into their images to control the model output for malicious purposes but they are unable to manipulate data provided by other users. In general, we consider a practical setting where the adversary serves as a single client in the supply chain, it only provides and controls data for the class it wants to attack. Therefore, the adversary can only select a subset of images *with the target label* to insert the trigger.

**The attacker's knowledge.** The adversary only has access to data for the target class that it provides. *No information of the victim model's architecture, the training process, or data from other clients* is exposed to the attacker.

## 4 Method

### 4.1 Problem Formulation

Let $f_\theta : \mathcal{X} \to \mathcal{Y}$ be a deep neural network that maps from an image $x \in \mathcal{X}$ to a label $y \in \mathcal{Y}$, and $\mathcal{D}_c = \{(x_1, y_1), \ldots, (x_n, y_n)\}$ be the clean training dataset. In backdoor attacks, the adversary first defines a trigger injecting function $T : \mathcal{X} \to \mathcal{X}$ that implants a trigger into an input and then applies $T$ to $m$ images in $\mathcal{D}_c$.

Let $S$ be the target class. The attacker selects a subset $S' \subset S$ of size $m$ and adds triggers to samples in $S'$. After injecting the trigger into $S'$ (and leaving the other examples in $S$ intact), the attacker gives its data to the victim and the victim combines that data with data from other sources to create a poisoned dataset $\mathcal{D}_p$. The victim then trains the model on $\mathcal{D}_p$ with the standard training pipeline to obtain the model $f_{\theta^*}$. The goal of the attacker is that any model that is trained on $\mathcal{D}_p$ would return correct predictions on unpoisoned examples but predict the target label $y^t$ on any example on which the trigger function $T(\cdot)$ is applied. Formally, for a benign input $x$ with the correct label $y$, we have

$$f_{\theta^*}(x) = y, \quad f_{\theta^*}(T(x)) = y^t.$$

The performance of backdoor attack methods is usually evaluated via two metrics: benign accuracy (BA) and attack success rate (ASR). BA is the accuracy of the infected model on benign test samples. ASR is the proportion of attacked test samples that are successfully predicted as the target label by the infected model. In addition, stealthiness is an important factor for backdoor attacks, which is reflected by small poisoning rate, imperceptibility of the backdoor, and resistance against backdoor defense methods.

## 4.2 Selecting Samples with Self-Supervised models

We start with a simple question: "*Why are dirty-label attacks more effective than clean-label attacks?*". The difference between them is the samples selected to insert the triggers. For dirty-label attacks, poisoned data comes from various labels, its features are dissimilar to those in the target class. For example, if the adversary wants to attack class $0$, dirty-label attacks can choose samples from class $0, 1, 2, \ldots$, while clean-label attacks only pick samples in class $0$.

During the training process, the model looks for common features to form the decision boundary. Therefore, an example containing features different from other examples in a class is harder to learn. When the adversary injects a trigger and alters the label, *the model can not rely on existing features in the image to optimize the objective function, instead it favors backdoor features*, leading to a higher attack success rate.

Based on that intuition, we search for and add triggers to hard samples in the target class to achieve stronger clean-label attacks. A straightforward solution is to train a surrogate model on the training set and examine the behavior of the model on each data point. For example, a sample with a higher loss value is likely to be more difficult to learn. However, this method violates our threat models as it requires information from other classes, and training a surrogate model is also computationally expensive.

We propose a simple strategy called **B**ackdoor **A**ttacks by **SE**lective Poisoning with **L**imited Information (BASEL) to find examples that are dissimilar to other data in the target class by exploiting pretrained self-supervised models. These samples are far from other samples in the feature space and can be identified by extracting feature representations, and then computing the distance between them. To obtain discriminative and task-agnostic features, we use self-supervised models. Let $g$ be a feature extractor, we define the distance between two samples $x_i, x_j$ by cosine similarity between their feature $z_i = g(x_i), z_j = g(x_j)$:

$$d(x_i, x_j) = \frac{z_i^\mathsf{T} z_j}{\|z_i\|\|z_j\|}.$$

---

**Algorithm 1** Sample selection algorithm

---

**Input:** a self-supervised model $g$,
    target class dataset $S$, attack budget $m$
**Output:** $S' \subset S$ where $|S'| = m$
    **for** $x_i \in S$ **do**
        $z_i \leftarrow g(x_i)$
    **end for**
    **for** $x_i \in S$ **do**
        Compute $s(x_i)$ by Equation 1
    **end for**
    $S' \leftarrow$ set of $m$ samples with the highest $s(x)$

---

We apply the classical $k$-NN algorithm to calculate a score function $s(x)$ as the mean of distances between $x$ and its $k$-nearest neighbors $x_1, \ldots, x_k$ in the target class in terms of the distance $d(\cdot, \cdot)$:

$$s(x) = \frac{1}{k} \sum_{i=1}^{k} d(x, x_i). \tag{1}$$

With an attack budget of $m$, our strategy collects $m$ samples with the highest scores. The detailed algorithm is shown in Algorithm 1.

# 5 Experiments

In this section, we empirically evaluate the performance of our selection strategy.

## 5.1 Experimental Setup

**Dataset.** We consider two widely used benchmark datasets: CIFAR10 [14] and GTSRB [28].

Table 2: The attack success rate (ASR) of clean-label attacks on CIFAR10 with $1\%, 5\%$ and $10\%$ of the target class being poisoned.

| Model | Strategy | BadNets | | | Blended | | | SIG | | |
|---|---|---|---|---|---|---|---|---|---|---|
| | | 1% | 5% | 10% | 1% | 5% | 10% | 1% | 5% | 10% |
| ResNet18 | Random | 10.27 | 12.60 | 17.21 | 21.00 | 38.81 | 48.36 | 39.65 | 67.33 | 71.26 |
| | BASEL | **11.58** | **19.82** | **28.72** | **31.81** | **62.40** | **75.88** | **71.81** | **86.25** | **91.66** |
| VGG19 | Random | 10.28 | 16.49 | 19.65 | 14.08 | 26.06 | 35.41 | 17.64 | 43.76 | 69.35 |
| | BASEL | **11.73** | **29.43** | **29.91** | **15.79** | **39.49** | **52.15** | **19.69** | **77.13** | **84.23** |

Table 3: The attack success rate (ASR) of clean-label attacks on GTSRB with $1\%, 5\%$ and $10\%$ of the target class being poisoned.

| Model | Strategy | BadNets | | | Blended | | | SIG | | |
|---|---|---|---|---|---|---|---|---|---|---|
| | | 1% | 5% | 10% | 1% | 5% | 10% | 1% | 5% | 10% |
| ResNet18 | Random | **6.05** | 6.12 | 6.55 | 33.24 | 53.78 | 62.53 | 43.12 | 57.72 | 63.77 |
| | BASEL | 5.89 | **6.85** | **7.40** | **44.29** | **55.07** | **67.09** | **43.46** | **59.02** | **67.09** |
| VGG19 | Random | 6.42 | 6.68 | 7.48 | 35.55 | 46.12 | 52.72 | 21.61 | 40.42 | 48.98 |
| | BASEL | **7.21** | **7.26** | **11.12** | **39.16** | **55.19** | **60.17** | **38.19** | **51.43** | **56.19** |

**Models.** For the victim model, we consider ResNet18 [11] and VGG19 [26].

**Attacks.** We adapt trigger patterns from BadNets, Blended and SIG.

**Strategy.** For comparison, we apply two strategies: the baseline random sample selection and BASEL with VICReg [2] as a feature extractor.

## 5.2 Effectiveness of BASEL

We perform clean-label attacks on CI-FAR10 with the random strategy and BASEL, and report the attack success rate in Table 2. As can be observed, our strategy outperforms the random baseline on all the attacks, models and poisoning rates. These results validate our intuition, showing there exist samples in the dataset that are more suitable to perform the attack. Although with very limited information, given the target class data only, BASEL still boosts the attack success rate of clean-label attacks significantly. For instance, with BadNets

Table 4: The clean accuracy (BA) of random strategy and BASEL on CIFAR10 and GTSRB with various poisoning rate.

| Model | Strategy | CIFAR10 | | | GTSRB | | |
|---|---|---|---|---|---|---|---|
| | | 1% | 5% | 10% | 1% | 5% | 10% |
| ResNet18 | Random | 95.24 | 94.94 | 95.11 | 94.37 | 93.46 | 94.90 |
| | BASEL | 95.15 | 95.17 | 95.12 | 94.32 | 94.19 | 93.75 |
| VGG19 | Random | 91.92 | 91.71 | 91.53 | 92.98 | 93.60 | 93.01 |
| | BASEL | 91.82 | 91.76 | 91.82 | 93.17 | 93.00 | 93.35 |

trigger, the random strategy reaches less than $20\%$ success rate and BASEL increases it by more than $10\%$. For Blended and SIG, BASEL consistently improves the attack success rate by a large margin on both models, up to $30\%$. Specifically, with just $10\%$ poisoning rate on the target class ($1\%$ of the whole dataset being poisoned), BASEL helps SIG achieve more than $90\%$ success rate when attacking ResNet18.

Furthermore, we study the effect of BASEL on GTSRB, which is a challenging dataset. It has 43 classes with imbalanced data, and more importantly, samples in different classes are less discriminative compared to CIFAR10. Table 3 illustrates that our method still boosts the performance remarkably. For instance, with $5\%$ poisoning rate of the target class, BASEL improves the success rate by around $10\%$ for Blended and SIG on VGG19.

Finally, we evaluate the effect of BASEL to the clean performance of the model. Table 4 exhibits the clean accuracy when applying random strategy and BASEL with SIG attack on CIFAR10 and GTSRB, implying that BASEL causes no degradation to the performance on benign data. The results for other triggers are similar.

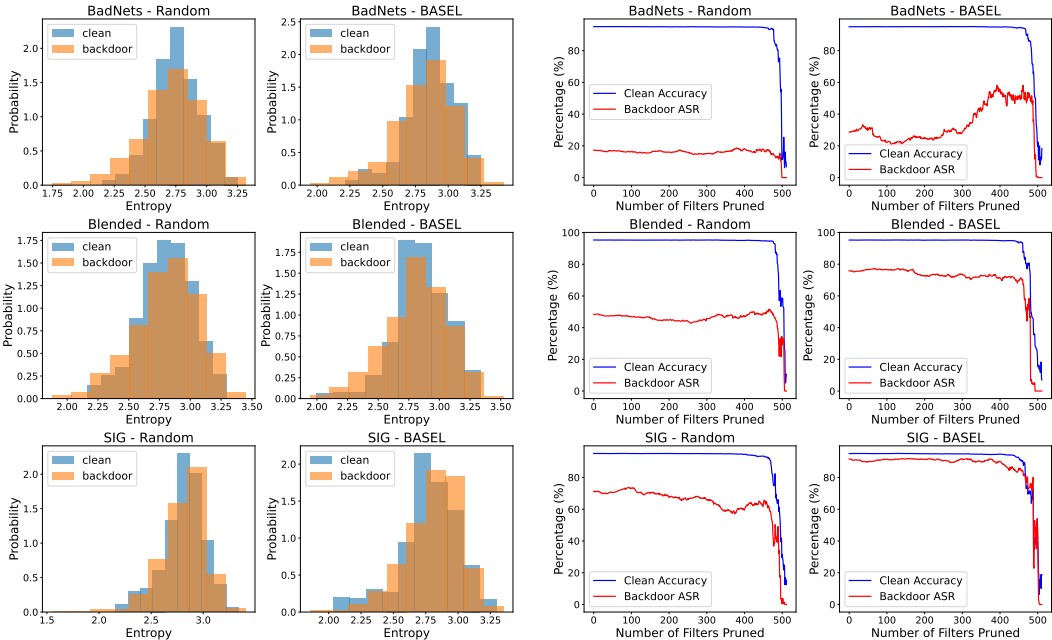

(a) Performance against backdoor detection method STRIP. The behavior of the infected model is similar between clean and backdoor data, showing the stealthy of BASEL under STRIP detector.

(b) Performance against backdoor mitigation method Fine-pruning. The plot of clean accuracy and attack success rate during filter pruning shows the resistance of BASEL to Fine-pruning.

Figure 1: Performance of BASEL against backdoor defenses

## 5.3 Performance against Backdoor Defenses

We evaluate our strategy with backdoor defense methods, which are STRIP [7] (backdoor detection) and fine-pruning [18](backdoor mitigation). We test these defenses on a ResNet18 model trained on CIFAR10 with $10\%$ of the target class being poisoned.

**STRIP.** This method is an inference-time defense by perturbing the input and examining the entropy of the output. A sample with low entropy is more likely to be poisoned. Figure 1a visualizes the entropy of the output of clean data and backdoor data with random strategy and BASEL. We observe that with BASEL, the behavior of the poisoned model is similar between clean and backdoor data, being stealthy under STRIP detector.

**Fine-pruning.** We evaluate the resistance of BASEL under Fine-pruning, which is a backdoor mitigation method. Given a benign sample, it assumes that inactivated neurons are responsible for backdoor features and gradually prunes these neurons. We plot the clean accuracy and attack success rate during this process in Figure 1b, showing that BASEL is resistant to Fine-pruning and consistently achieves higher ASR than the random strategy.

## 6 Conclusion

In this work, we study the sample selection problem in clean-label backdoor attacks. We propose a selection strategy that works with a single class data only by exploiting self-supervised models to select hard samples. Empirical results show that our method increases the attack success rate of clean-label attacks significantly. Furthermore, our strategy is resistant to backdoor defenses. Finally, it will be interesting to study the combination of our method and other preprocessing methods that make samples harder to learn, and extend our strategy to dirty-label attacks in future works.

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

# Appendix

## A  Experimental Setup

### A.1  Dataset

We conduct experiments on two widely used benchmark datasets: CIFAR10 [14] and GTSRB [28].

**CIFAR10** contains images from 10 classes, with $50,000$ samples for the training set and $10,000$ samples for the test set.

**GTSRB** is a imbalanced dataset with 43 classes of traffic sign images, including $39,209$ samples for training and $12,630$ samples for test.

### A.2  Models

For the victim model, we consider ResNet18 [11] and VGG19 [26]. They are trained with AdamW [20] optimizer for 300 epochs with cosine learning rate schedule. The initial learning rate is set to $5e-4$ for ResNet18 and $1e-4$ for VGG19.

### A.3  Attacks

We adapt two trigger patterns from dirty-label attacks, which are BadNets and Blended. For BadNets, a checkerboard pattern [29] is added to the image. For Blended, we implant a Hello Kitty image with the blended rate $\alpha = 0.2$. Also, we evaluate our strategy on SIG, a clean-label attack, with $\Delta = 20$ and $f = 6$. We perform the clean-label attacks to class 0 for CIFAR10 and class 1 for GTSRB. These attacks inject triggers to $1\%, 5\%$ and $10\%$ of the target class, which are $0.1\%, 0.5\%, 1\%$ poisoning rate with respect to the whole dataset in CIFAR10, and $0.04\%, 0.19\%, 0.38\%$ in GTSRB.

### A.4  Strategy

For comparison, we apply two strategies: the baseline random sample selection and BASEL. To get the vector representation for BASEL, we employ VICReg [2], a self-supervised model pretrained on ImageNet [5] with ResNet50 as the architecture.

