# OpenReview forum: "Clean-label Backdoor Attacks by Selectively Poisoning with Limited Information from Target Class"
_NeurIPS.cc/2023/Workshop/BUGS — NeurIPS 2023 BUGS Poster_

### Official Review · Reviewer_ekEu · 2023-10-20
**An interesting work that needs more detail and experiments**

**Rating:** 6
**Confidence:** 4

**Review:**

This paper further explores how to select critical (instead of random) samples for poisoning to improve the effectiveness of clean-label backdoor attacks. In general, they used a pre-trained self-supervised model to obtain the features of all training samples and find outliers (with a large average distance to their k-nearest samples) as the hard samples for poisoning. The authors evaluate their method on CIFAR and GTSRB datasets and also discuss its resistance to two backdoor defenses.

Pros
1. The paper is well written and its idea is easy to follow.
2. The topic is of great significance and interest to the audiences in this workshop.
3. This method not requires model training, which is efficient.

Cons
1. Please provide a real-world case that the attackers can only provide samples from the single class.
2. The authors should compare their method with existing works (e.g., [8] and [31]).
3. I would like to see more discussions about its effectiveness when the domain of pre-trained self-supervised model is (significantly) different from the one of the training dataset.

---

### Official Review · Reviewer_Kg9d · 2023-10-27

**Rating:** 4
**Confidence:** 5

**Review:**

The paper proposes to smartly select samples for poisoning for clean label attack. The intuition is that those samples must be hard for the model to learn, and hence the proposed approach is to select hard sample. The manner in which the hard samples are collected is finding the points that have the highest avg distances from their KNN in the representation space of a trained self-supervised model.

The authors demonstrate the effectiveness of their attack on two datasets, CIFAR10 and GTSRB, using poisoning ratios of 1, 5, and 10%, and ASR and accuracy as the metrics. The baseline is the random selection of datapoints from the target class, and show that the proposed method BASEL gets much higher ASR for the same poisoning percentage compared to the random selection baseline.

Problems with the paper:
1. The authors state that the adversary only has access to a subset of the data, but in the experiments they use the self-supervised model to get representations of all the datapoints of the target class, and then selectively poison the hardest samples according to their criteria. In real world, the attacker will not have access to thousands of datapoints to compare from and select the hardest ones, they can have some samples, but not many. This experimental setup violates the assumption (which is a perfectly valid argument) the authors make.
2. And continuing on the same thread, a relevant baseline would the the first work that did clean label attack, cited [29] by the authors. Random is not the best baseline for this work.
3. The last point about the unrealisticness of the attack is the high poisoning percentages. Even poisoning 1% of the dataset is very hard for large scale datasets. See the work by Carlini where they poison less than 0.01% (https://arxiv.org/abs/2106.09667) of the dataset to get almost 95% ASR for multimodal models, that is a more realistic poisoning percentage. However, I do understand that several past works have reported number upto 10% data poisoning which I personally believe is unrealistic.

---

### Decision · Program_Chairs · 2023-10-28

**Decision:**

Accept (Poster)

**Comment:**

Both reviewers have opinions on the baseline methods, however, considering this is a workshop submission, we believe the paper has value to the community. We have decided to accept this paper. However, the authors are encouraged to improve their baseline methods to make evaluations more comprehensive.